# Understanding China's National Emergency Command System from the Perspective of Power and Responsibility Allocation

Feng Kong [1,2]

1   College of Humanities and Development Studies, China Agricultural University, Beijing 100083, China; kongfeng0824@cau.edu.cn
2   Center for Crisis Management Research, Tsinghua University, Beijing 100084, China

**Abstract:** The structure between the Party and the government is the core and soul of China's emergency management, and the allocation of power and responsibility is the core of China's national emergency command system (NECS). The allocation of power and responsibility between the Party and the government, as well as between departments, is the main aspect of the allocation of power and responsibility in China, and is also an important component of the NECS. This paper mainly introduces the characteristics of power and responsibility allocation between the Party and the government, as well as between departments in China's NECS, and analyzes the above-mentioned power and responsibility allocation, based on the prevention and control of SARS (severe acute respiratory syndrome) and COVID-19 (coronavirus disease 2019), and analyzes their development, changes, and unchanged characteristics. Through the above analysis, we found that the Party's leadership style in dealing with emergencies has changed from indirect leadership to direct leadership. The joint defense and control mechanism has replaced the national headquarters of emergency management as the common mode.

**Keywords:** national emergency; command system; power; responsibility allocation; China; emergency management

## 1. Characteristics of China's NECS

China's national emergency management system is composed of various levels and types of substantive institutions [1,2]. China's national emergency management institutions mainly include a leading organization, administrative agencies, local institutions, expert groups, etc. [3]. Among them, the leading organization include not only the Party committees and governments at all levels, but also all kinds of deliberative and coordinating organization. The national emergency command system (NECS) is an organization and leadership mode for China in response to emergencies; it is a leading organization at the top level that plays the most important role in organization decision-making, command, and dispatch [1,3,4]. According to the degree of social harm, the scope of influence, and other factors, emergencies can be divided into four levels in China: particularly serious, major, large, and general. The grading standards for emergencies are generally formulated by the State Council. When encountering extraordinary emergencies, such as large-scale natural disaster, extraordinary public health events, mega mass incidents, and catastrophic accidents, the State Council establishes an NECS under the leadership of the Communist Party of China (CPC) Central Committee and in accordance with the Emergency Response Law of the People's Republic of China [1]. In the past few decades, China has established the NECS in many major emergencies, such as the national catastrophic flood in 1997, SARS (severe acute respiratory syndrome) in 2003, the Wenchuan earthquake in 2008, the influenza A epidemic in 2009, and COVID-19 (coronavirus disease 2019) in 2020. With the economic and social development, China's NECS has been constantly adjusted and changed [1,5,6], especially during the last 20 years. In particularly, the role of the Party has changed from indirect leadership to direct leadership. Compared with the emergency

command department, the cross-department joint defense and control mechanism is more popular and adopted by the government.

## 2. Structure of Power and Responsibility Allocation in China's NECS

The core of China's NECS is the allocation of power and responsibility. Different power systems, such as legislation, administration, justice, and the armed forces, play a certain role in dealing with emergencies in various countries [4]. The power and responsibility allocation in China's emergency response activities mainly includes two levels: the upper level includes the power and responsibility allocation between Party committees and governments in different power systems, which can be called an inter-system relationship; the lower level is the power and responsibility allocation between different government departments, which can be called an inter-departmental relationship (Figure 1). The understanding of China's NECS can be carried out from the power and responsibility allocation between the Party and the government, as well as between government departments.

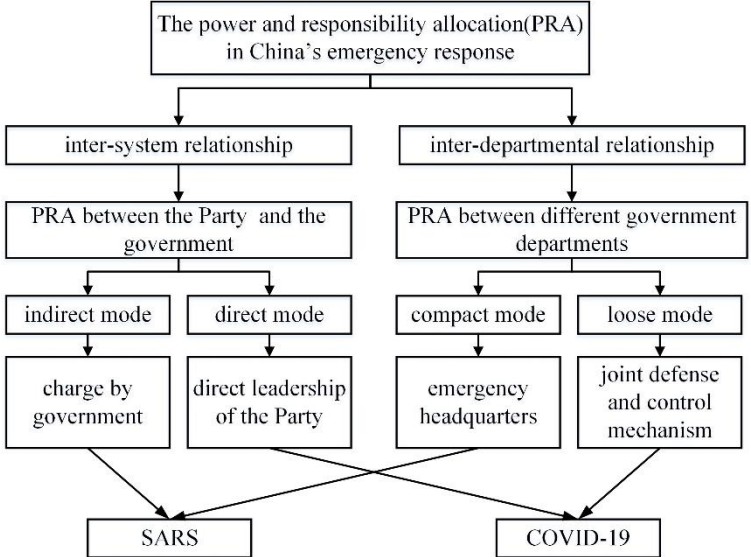

**Figure 1.** Power and responsibility allocation of China's national emergency command system (NECS).

## 3. Power and Responsibility Allocation Between the Party and the Government in China's NECS

As a special political force, the CPC occupies a leading position in China's national life, and the relationship between the Party and the government is also the most basic political relationship in China. China's unique political system can be summarized as the separation of six state powers—namely, the leadership of the Party Central Committee, the legislative power of the National People's Congress, the executive power of the State Council, the consultative power of the Chinese People's Political Consultative Conference, the judicial power of the Supreme People's Court and the Supreme People's Procuratorate, and the military power of the Central Military Commission [3,7]. In the structure of China's power system, the six state powers mentioned above are based on the division of functions; among them, the Party's leadership and integration is the basis of the division of powers system. Through strong political leadership, the Party has profoundly shaped and integrated into the government at the organizational and ideological levels, forming a centralized and unified Party and government structure. Government agencies at all levels also carry out relevant administrative work under the leadership of Party committees at all levels. The effective operation of the Party and the government governance structure with Chinese characteristics is considered to be one of the important reasons for China's economic and social development to achieve remarkable performance.

In the field of emergency response, the Party and the government governance structure with Chinese characteristics also plays a very important role [6]. Direct leadership and indirect leadership can be divided into two types, according to different ways of leadership. According to the degree and function of the Party in dealing with emergencies, the power and responsibility allocation between the Party and the government in the NECS can be divided into two categories: indirect mode and direct mode (Figure 1). Indirect mode means that the emergency response is mainly determined as the work within the scope of the government's functions and powers, which belongs to administrative affairs. The government is in charge of the emergency command, and the Party plays an overall role behind the scenes and does not directly command. Direct mode refers to when the Party moves from behind the scenes to the front of the stage, and is directly responsible for the organization and command of emergency response activities. In this situation, the government is responsible for implementing major decision-making determined by the Party committee under the direct leadership of the Party. Generally, the indirect mode of NECS is a system of indirect leadership of the Party committee and direct leadership of the government. Its basic operation mode is that the government takes the lead in the emergency command headquarters or joint defense, as well as the control mechanism and other forms of emergency command structure, undertaking the responsibility of the emergency response. The Party committee is also mainly responsible for guiding, coordinating, and making decisions on major issues. Different from the indirect model, the direct model of the NECS is a system under the direct leadership of the Party committee and the implementation of the government. Its basic operation mode is that the Party committee directly establishes an emergency command department or leading group to be responsible for the decision-making of major events. The leader of the emergency command department or leading group is the person in charge of the Party committee, or the person in charge of the Party committee and the government at the same time. Under the leadership of the Party committee, the government establishes a certain type of emergency command structure, which is responsible for implementing the major decisions and arrangements made by the Party committee's emergency command headquarters or leading group leaders, and executing overall coordination for the emergency response.

## 4. Power and Responsibility Allocation between Different Departments in China's NECS

Whether it is the direct mode or the indirect mode of NECS, emergency response is ultimately undertaken by the Party committees and government at all levels in China. The Party committees and government at all levels are composed of different departments. The relationship between different government departments is called an inter-departmental relationship. The relationship between government departments describes the level structure of the organization, and the complexity of the relationship between government departments is much greater than that of the central government. The emergencies that need the leadership of state-level organizations to deal with are generally cross-border major emergencies. These events often span different regions and departments in terms of the causes, development processes, and impact consequences, which often produce a chain reaction. In the face of cross-border emergencies, it is necessary to establish a certain form of cross-border emergency command organization, integrate the resources of relevant departments of the Party committee and the government, and carry out a coordinated response. This kind of cross-department emergency command organization reflects the power and responsibility allocation among different departments for emergency response. To understand the constitution of China's NECS, we should not only pay attention to the power and responsibility allocation between the Party and the government, but also investigate the power and responsibility allocation between different departments.

According to the degree of the relationship between departments, China's NECS can be divided into two categories: the compact model and loose model (Figure 1). The compact model refers to the establishment of emergency command organization led by the government leader, integrating the strength and resources of relevant departments,

forming a management mode with the basic characteristics of "issuing orders/obeying execution orders". A loose model refers to a kind of mutual consultation and participation management mode formed by cooperation between relevant departments based on volunteerism and trust. The compact model of NECS is represented by the emergency headquarters or the emergency leading group. Its basic operation characteristics are that the government takes the lead to set up an emergency headquarters or an emergency leading group to undertake the responsibility of responding to emergencies; the head of the government is the chief commander of the emergency headquarters, and several working groups are set up under the unified leadership and command of the emergency headquarters. The relevant departments are integrated into the emergency headquarters to carry out unified action under the unified leadership and command. The loose model of NECS is represented by the joint defense and control mechanism in China. Its basic operation characteristics are that the main department responsible for emergency response takes the lead to establish an inter-departmental joint defense and control working mechanism, and relevant departments participate as member agencies; under the joint defense and control working mechanism, there are several working groups, with the heads of relevant departments as the team leaders to clarify the responsibilities of each department, the member agencies' work, and cooperation with one another.

## 5. China's NECS in Response to SARS and COVID-19

The NECS for fighting against SARS in 2003 was a very typical mode of indirect leadership by the Party committee and unified command by the government (Figure 1). The CPC Central Committee played an indirect leadership role by holding meetings, listening to reports, and making major decisions, while the government was specifically responsible for the organization and command of epidemic prevention and control. The mode of unified command and dispatch was adopted by the emergency headquarters. The national SARS prevention and control headquarters was composed of personnel from more than 30 departments of the CPC Central Committee, the State Council, the military system, and Beijing Municipality. The vice Premier in charge of the State Council was the general commander, and the emergency office was located in the General Office of the State Council.

The NECS for fighting against COVID-19 is a typical mode of direct leadership by the Party committee, with joint defense and control by the government at all levels (Figure 1). The CPC Central Committee has set up an emergency leading group to deal with the epidemic and make decisions directly. The government is mainly responsible for implementing the Party committee's decision-making and carrying out coordination and scheduling. The joint defense and control mechanism of the State Council has been established to coordinate the work of epidemic response.

There are two aspects of the NECS that have not changed in the above two epidemic responses. The first is that the Party has always played a leading core role in emergency response. The second is continuously strengthening cross-departmental coordination and linkage. The emergency headquarters and the inter-departmental joint defense and control mechanism are two different specific working modes. However, no matter which form is adopted, the purpose is to strengthen the coordination and cooperation between departments.

China's NECS has changed in two aspects in the above two epidemic responses. First is that the Party's leadership style in dealing with emergencies has changed from indirect leadership to direct leadership. In terms of the allocation of power and responsibility of the Party and the government, the government led the fight against SARS in 2003, and the Party played an indirect leading role by holding meetings, listening to reports, and making major decisions and arrangements. In the process of coping with COVID-19 in 2020, the CPC Central Committee directly conducted emergency command on the prevention and control of epidemic situation. Second, the NECS is more inclined to adopt cross-department joint defense and control mechanisms, and start fewer emergency headquarters. From

the perspective of the allocation of power and responsibility between departments, different from the establishment of the national command for SARS prevention and control in 2003, an inter-departmental joint defense and control mechanism was set up in response to COVID-19 in 2020, with the main responsible department as the convener and the relevant departments as the member agencies to coordinate the cross departmental emergency response.

The change of China's NECS is a process of gradual adjustment. It is closely related to the occurrence of major emergencies, which belong to event-driven system construction, and is the development process of problem-oriented emergency management. China's national emergency command system has continuously improved and accumulated experience in the practice of major emergencies, so that it can reach the best level of emergency rescue under the allocation of rights and responsibilities in China. In addition, on the one hand, the improvement of China's comprehensive national strength has laid an economic foundation for the development and change of NECS. On the other hand, the Party and the government attach great importance to the innovation and development of NECS. Especially in recent years, China is constantly strengthening the construction of the Party's leadership, and most of the government decision makers are the Party members. Through the construction of inner-Party supervision and accountability, it is conducive to promote an effective response to major emergencies.

With the comprehensive strengthening of the Party's leadership and further deepening the reform of the Party and state institutions, it is expected that the NECS of direct leadership by the Party committee and joint defense and control by government departments at all levels, which is similar to COVID-19 response, will be more common in the future.

## 6. Conclusions

China's NECS reflects the organizational structure in emergency management, and at its core is the allocation of power and responsibility. The power and responsibility allocation in China's NECS includes an upper level and a lower level. The upper and the lower levels refer to the power and responsibility allocation between Party and governments, as well as the power and responsibility allocation between different government departments, respectively. Based on the allocation of power and responsibility, this paper analyzes the main characteristics and developmental trend of China's NECS among the party and governments, as well as the different government departments. The Party has always played a leading core role in emergency response in China, and the cross-departmental coordination and linkage has been continuously strengthened. Therefore, the Party's leadership style in dealing with emergencies has changed from indirect leadership to direct leadership. The emergency management mode of cross-department joint defense and control mechanism is more easily adopted during major public emergencies.

**Funding:** This research was funded by the National Basic Research Program of China, grant number 2018YFC0806900, the National Natural Science Foundation of China, grant number 41701103, 41775078, 41801064 and 71790611, and the Beijing Social Science Foundation Project, grant number 19JDGLA008.

**Institutional Review Board Statement:** Not applicable.

**Informed Consent Statement:** Not applicable.

**Data Availability Statement:** No new data were created or analyzed in this study. Data sharing is not applicable to this article.

**Acknowledgments:** The authors would like to acknowledge the helpful comments of two anonymous referees of the journal, who have helped to improve this paper. This research was funded by the National Basic Research Program of China (2018YFC0806900), the National Natural Science Foundation of China (41701103, 41775078, 41801064, 71790611), and the Beijing Social Science Foundation Project (19JDGLA008).

**Conflicts of Interest:** The author declares that no conflict of interest exists.

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
