# Peer review of "Understanding China’s National Emergency Command System from the Perspective of Power and Responsibility Allocation"

_sustainability, doi:10.3390/su13010301_

Round 1

Reviewer 1 Report

Dear authors
In this article, you touch upon a very time-sensitive topic, which is widely described in various journals.

There are some small shortcomings in your article:\n-
line 9 - the bilges are removed in bold "The" text.
Line 17 - missing space between COVID-19(coronavirus disease 2019).
Line 72 - missing the space between the very important role(Lv, 2017).
Line 119 - there is no space between the loose model(Figure.1). T
There is a lack of constructive conclusions from the situation described.

Author Response

Response to the Reviewer Comments

First of all, I would like to express my appreciation for the very insightful reviews, and my thanks to the reviewer for his/her interests in my work and efforts to help improve it. I revised the paper according the reviewer’s suggestion point by point, and marked it in red.

Dear authors

In this article, you touch upon a very time-sensitive topic, which is widely described in various journals.

There are some small shortcomings in your article:\n-

Line 9 - the bilges are removed in bold "The" text.

Answer: The reviewer’s suggestion is right. I have removed in bold "The" text.

Line 17 - missing space between COVID-19(coronavirus disease 2019).

Answer: The reviewer’s suggestion is right. I have added the space between COVID-19 and (coronavirus disease 2019).

Line 72 - missing the space between the very important role(Lv, 2017).

Answer: The reviewer’s suggestion is right. I have added the space between the very important role and (Lv, 2017).

Line 119 - there is no space between the loose model(Figure.1).

Answer: The reviewer’s suggestion is right. I have added the space between the loose model and (Figure.1).

There is a lack of constructive conclusions from the situation described.

Answer: The reviewer’s suggestion is right. I have added a constructive conclusions in the end of the maintext and marked it in red.

Reviewer 2 Report

The paper offers a clear description of how responsibility is allocated in the management of the emergency in China. An application to the cases of SARS and COVID-19 is also provided. The main paper's argument is that a paradigm shift is occurring from an indirect to a direct leadership approach. 

I think the paper offers some interesting insights into an emergency management system is not easy to have information on (that is, the China's one). Anyway, beyond the description of the system's organization and operating mechanisms, I think that the main argument of the paper should be stressed a bit more (for instance, the abstract and the introduction should mention the fact that the system is moving from one approach to the other). 

Author Response

Response to the Reviewer Comments

First of all, I would like to express my appreciation for the very insightful reviews, and my thanks to the reviewer for his/her interests in my work and efforts to help improve it. I revised the paper according the reviewer’s suggestion point by point, and marked it in red.

The paper offers a clear description of how responsibility is allocated in the management of the emergency in China. An application to the cases of SARS and COVID-19 is also provided. The main paper's argument is that a paradigm shift is occurring from an indirect to a direct leadership approach.

I think the paper offers some interesting insights into an emergency management system is not easy to have information on (that is, the China's one). Anyway, beyond the description of the system's organization and operating mechanisms, I think that the main argument of the paper should be stressed a bit more (for instance, the abstract and the introduction should mention the fact that the system is moving from one approach to the other).

Answer: The reviewer’s suggestion is right. I have added the fact that the emergency management system in China is moving from one approach to the other, and I marked it in red.
